# Temporal Evolution of PAHs Bioaccessibility in an Aged-Contaminated Soil during the Growth of Two *Fabaceae*

**DOI:** 10.3390/ijerph17114016

**Published:** 2020-06-05

**Authors:** Marie Davin, Elisa Renard, Kévin Lefébure, Marie-Laure Fauconnier, Gilles Colinet

**Affiliations:** 1Soil-Water-Plant Exchanges, University of Liège, Gembloux Agro-Bio Tech, 2 Passage des Déportés, 5030 Gembloux, Belgium; elisa-renard@outlook.be (E.R.); kevin.lefebure@uliege.be (K.L.); gilles.colinet@uliege.be (G.C.); 2Laboratory of Chemistry of Natural Molecules, University of Liège, Gembloux Agro-Bio Tech, 2 Passage des Déportés, 5030 Gembloux, Belgium; marie-laure.fauconnier@uliege.be

**Keywords:** polycyclic aromatic hydrocarbons (PAHs), bioaccessibility, rhizoremediation, aged soil, *Medicago sativa* L., *Trifolium pratense* L.

## Abstract

Polycyclic aromatic hydrocarbons (PAHs) are health-concerning organic compounds that accumulate in the environment. Bioremediation and phytoremediation are studied to develop eco-friendly remediation techniques. In this study, the effects of two plants (*Medicago sativa* L. and *Trifolium*
*pratense* L.) on the PAHs’ bioaccessibility in an aged-contaminated soil throughout a long-term rhizoremediation trial was investigated. A bioaccessibility measurement protocol, using Tenax^®^ beads, was adapted to the studied soil. The aged-contaminated soil was cultured with each plant type and compared to unplanted soil. The bioaccessible and residual PAH contents were quantified after 3, 6 and 12 months. The PAHs’ desorption kinetics were established for 15 PAHs and described by a site distribution model. A common Tenax^®^ extraction time (24 h) was established as a comparison basis for PAHs bioaccessibility. The rhizoremediation results show that *M. sativa* developed better than *T. pratense* on the contaminated soil. When plants were absent (control) or small (*T. pratense*), the global PAHs’ residual contents dissipated from the rhizosphere to 8% and 10% of the total initial content, respectively. However, in the presence of *M. sativa*, dissipation after 12 months was only 50% of the total initial content. Finally, the PAHs’ bioaccessible content increased more significantly in the absence of plants. This one-year trial brought no evidence that the presence of *M. sativa* or *T. pratense* on this tested aged-contaminated soil was beneficial in the PAH remediation process, compared to unplanted soil.

## 1. Introduction

Polycyclic aromatic hydrocarbons (PAHs) are persistent organic compounds of hydrophobic nature that are composed of fused rings in angular, linear or clustered arrangements [1]. PAHs mainly form during incomplete combustion, which is frequent in natural phenomena (volcanic eruption, forest fires) but also anthropogenic activities (car exhaustion, waste burning or domestic and industrial activities) [2]. Because of their potential (geno)toxicity and heavy presence in former industrial areas [3], PAHs have been the center of many remediation studies over the past decades.

On the one hand, many researchers have focused on PAH biodegradation pathways, which have been thoroughly reviewed [1,2,4]. When it comes to the environmental influence on hydrophobic organic compounds’ biodegradation, it is now well-known that the most important limiting factor is their bioaccessibility [5], i.e., the availability of a chemical to “cross an organism’s cellular membrane from the environment, if the organism has access to the chemical”, as defined by Semple et al. [6]. The word “bioavailability” is extensively used in the literature but, by definition, the bioavailable fraction only refers to the chemical that is available “to cross an organism’s cellular membrane from the environment at a given time”, so the term “bioaccessibility” will be preferred in this paper. Indeed, for biodegradation to take place, the targeted pollutants must come into contact with the degrading microorganisms or their enzymes. This mostly takes place in the aqueous soil solution [5]. However, hydrophobic compounds, such as PAHs, are prone to ageing. Such phenomena are caused by environmental components (such as soil organic or mineral matter) that physically or chemically segregate compounds, thus lowering their presence in the aqueous solution and lowering their accessibility to the degrading agents. Ageing happens through two main mechanisms, sorption and diffusion, that have been extensively studied and reviewed [7]. The concept of bioaccessibility, compared to bioavailability, suggests that even a compound bound to a soil particle can become available to an organism if it is released into the organism’s environment [8]. This is extremely important in the context of soil remediation because it means that treatments could influence the bioaccessible fraction of a pollutant.

On the other hand, two types of environmentally friendly remediation technologies are being developed, bioremediation and phytoremediation, which rely on the use of living microorganisms or plants to remediate pollutions. Even if they tend to be referred to as different technologies, they cannot be considered separately when applied in soil as close interactions exist between plants and microorganisms in all the soil’s compartments (solid, liquid, and gas). The use of these interactions as a way to enhance the PAHs’ biodegradation is named rhizoremediation and is based on the observation that the rhizosphere creates favorable chemical and physical conditions for the soil microbiota to thrive [9]. It has been hypothesized that this rhizospheric effect is a combination of physical and chemical positive effects, as roots are believed to: (i) facilitate contact between soil particles and the microbiota [10]; (ii) increase soil oxygenation, which initiates aerobic metabolic pathways; (iii) exudate sugars, amino acids and organic acids, which serve as sources of energy for the microbiota [11]; (iv) release secondary metabolites with structural analogy to PAHs, which could induce microbial catabolic genes and co-metabolism [9]; and (v) enhance pollutants’ bioaccessibility. Indeed, some studies suggest that some compounds exuded by roots could desorb hydrocarbons from soil particles [12]. Besides, secondary metabolites are a collection of structurally different compounds (terpenes, nitrogen-containing products, phenolic compounds) among which some exhibit tensioactive properties [13]. For example, saponins are a diverse group of molecules composed of non-sugar aglycones coupled to sugar chain units, which gives them surface-active properties [14]. In a previous study [15], we hypothesized that exudates from two *Fabaceae* (*Medicago sativa* L. or *Trifolium pratense* L.) could, as part of the rhizospheric effect, enhance the PAHs’ bioaccessibility in an aged-contaminated soil, and thus enhance the PAHs’ biodegradation. The results showed that a single-dose addition of root exudates to an aged-contaminated soil in a microcosm incubation experiment (4 weeks), did not enhance the PAHs’ bioaccessibility nor dissipation. Given that plant root exudates are released at continuous rates into the environment [16], the following study was designed as a way to evaluate whether the prolonged presence of living *Medicago sativa* L. or *Trifolium pratense* L. on PAH aged-contaminated soil could enhance the PAHs’ bioaccessibility and, hence, facilitate their dissipation.

The tested plants were chosen for the following reasons. (i) Due to their symbiotic relationship with nitrogen-fixating bacteria, *Fabaceae* members have a better potential to grow on disturbed soils that often present unfavorable conditions to plant growth. Therefore, the most common *Fabaceae* genera (such as *Medicago* sp or *Trifolium* sp) are encountered on various terrestrial environments, and very often in open and disturbed land [17]. (ii) Saponins are present in a large variety of plants, including members of the *Fabaceae* family [18]. (iii) They have already been highlighted as good phytoremediation candidates [17] through other phytoremediation studies; thus, this experiment could bring original insight to the mechanisms at work.

As the main objective of the study was to assess the PAHs’ bioaccessibility in an aged-contaminated soil throughout a long-term rhizoremediation trial, the first step was to adapt a comparative bioaccessibility measuring protocol (using Tenax^®^ beads) to the experimental soil. Therefore, PAH desorption kinetics were measured for the soil and modelled in order to assess a common extraction time for all PAHs. The second step was then to apply the protocol to measure the PAHs’ bioaccessibility in an aged-contaminated soil that had been in the presence of *Medicago sativa* L. or *Trifolium pratense* L., for 3, 6 or 12 months, compared to unplanted soil. The residual PAH contents were also measured in soil at the end of each culture period.

## 2. Materials and Methods

### 2.1. Soil Material

The aged-contaminated soil used for this study was sampled on a brownfield (Marchienne-au-Pont, Belgium). The coordinates are 50°24′51.4″ N 4°24′39.1″ E. The site hosted a steel company from 1863 to 2012 and has been exposed to PAHs and trace elements. Soil was sampled, sieved through an 8 mm sieve, allowed to dry in ambient air, and stored in sealed boxes until further use. Before the experiments, the contents of 15 PAHs were determined for a total of 917 ± 146 µg g^−1^ DW (Table 1). These PAHs are part of the 16 PAHs on the American Environmental Protection Agency (EPA) watch list. The sixteenth PAH compound (acenaphtylene) was not detected in the experimental soil. From now on, the term “total PAHs” will designate the 15 PAHs detailed in Table 1. PAHs were also grouped in categories: ∑2–3 rings or light molecular weight PAHs of three rings or less (N, Ace, Fle, Phen and Anthr), ∑4 rings or intermediate molecular weight PAHs of four rings (F and Pyr), ∑4–6 rings or heavy molecular weight PAHs of four rings or more (BaA; Chrys, BbF, BkF, BaP, DBahA, BghiP and IcdP), and ∑all or total PAHs (N to IcdP) [19]. The soil was also presented with metal contamination (541, 171, 1.39, 357, and 3373 µg g^−1^ DW of Cr, Cu, Hg, Pb, and Zn, respectively) but not petroleum hydrocarbons, PCBs, or BTEXs. The particle size distribution (75% sand, 19% silt, 6% clay) identified the soil as loamy sand, the ${\mathrm{pH}}_{H_{2}O}$ was 10.0, and the total organic carbon was 18.9 ± 0.22% (*w*/*w*). These last two parameters were very high compared to values encountered in uncontaminated soils.

### 2.2. PAHs Bioaccessibility Measurement

The PAHs’ bioaccessibility measurement protocol was developed based on a modelling technique previously described and used on a different aged-contaminated soil [15] but it will be reminded hereafter. The objective was to determine the time of contact between the soil solution and the Tenax^®^ beads (which serve as surrogate for the soil microbiota) that would extract the bioaccessible fraction of PAHs in the aged-contaminated soil.

#### 2.2.1. PAHs Desorption Kinetics

In order to compare the PAHs’ bioaccessibility throughout time and after different treatments, a comparison protocol was adapted from Cornelissen et al. [20] and Barnier et al. [21]; then a specific extraction time, representative of the bioaccessible fraction, was determined for the studied soil. First, the desorption kinetics of all PAHs in the studied soil were measured five times: 2.0 g of soil were weighed into glass centrifuge tubes with 0.5 g of Tenax^®^ beads (60–80 mesh) and 50 mL of an aqueous solution (0.01 M CaCl_2_ and 0.003 M NaN_3_ as biocides to prevent PAH degradation). The tubes were agitated for 1, 2, 4, 8, 16, 24, 48, 72 or 96 h on a rotary device (40 cycles min^−1^) and centrifuged (10 min; 2000× *g*) to separate the Tenax^®^ beads from the soil. The floating beads were collected by vacuum filtration and sorbed PAHs were extracted from Tenax^®^ beads by three repetitions of a 60 min sonication in presence of 20 mL of a 50:50 (*v*/*v*) n-hexane: acetone mixture. The combined organic phases were replaced with acetonitrile using a rotative evaporation device. The final acetonitrile extract was weighed for volume determination and analyzed for PAHs. 

After this, the remaining PAH sorbed fractions in the soil were calculated as follows:(1)$$\frac{S_{t}}{S_{0}}=\frac{C_{tot_{in}}-C_{ext_{t}}}{C_{tot_{in}}}$$
where $C_{tot_{in}}$. is the total initial PAH concentration in the soil [µg g^−1^ DW]; $C_{ext_{t}}$ is the amount of PAH extracted by Tenax^®^ beads after t hours of contact [µg g^−1^ DW]; $S_{t}$ is the sorbed fraction of compound remaining after t hours of extraction; and $S_{0}$ is the initial sorbed fraction, assumed to be the total initial PAH concentration.

#### 2.2.2. PAHs Desorption Modelling

Several desorption models were tested to describe the PAHs desorption data (Table 2). Models were generated using R 3.4.3. and the following packages: “minpack.lm”, “AICcmodavg”, and “plotrix”. The Levenberg-Marquardt algorithm was used to minimize squared residuals between the experimental and calculated values [22]. The Bayesian information criterion (BIC) was also calculated to select the best model for each PAH as follows:(2)BIC=k.ln(n)−2.ln(L)
where *k* is the number of parameters of a model, *n* is the number of data points, and *L* is the maximized value of a likelihood function. The R function is BIC (model_iner2).

#### 2.2.3. PAHs Desorption Parameters

The best models describing the PAH desorption kinetics were used to determine a common extraction time (t_ex_) for bioaccessibility measurement, which is the time for the most accessible PAH fraction to equilibrate with Tenax^®^ beads. In the models, it represents the time in which the slope closes down to zero. The slope limit was arbitrarily set to 10^−3^ and successive approximations were made according to the following equation:(3)$$\frac{y_{t_{ex}-24}-y_{t_{ex}}}{24}\leq 0.001.$$
where *y* is the calculated value of a PAH desorption equation at different times and *t_ex_* is the extraction time [h].

The highest of all the calculated *t_ex_* was kept in the common comparative measuring protocol and used in the rhizoremediation experiment.

### 2.3. Rhizoremediation Experiment

The rhizoremediation experiments were conducted in pots placed outdoors. Neither temperature nor sunshine time were controlled. Forty-five pots of dimension 10 × 10 × 15 cm each received 1 kg of dry experimental soil. Thirty pots were seeded with either 25 kg ha^−1^ (20 seeds per pot) of *Medicago sativa* L. (MS) or *Trifolium pratense* L. (TP) and 15 control samples (C) were left unplanted. Seeds were tested prior to the experiment and had a 100% germination rate. The pots were placed outdoors and arranged in a completely randomized block. The experiment lasted from April 2018 to April 2019, so that the plants would be exposed to a year of weather changes. During that year, the nearby weather station registered several drought episodes, a total of 169 dry days and 615 mm of cumulated precipitation instead of the normal 823 mm of this area (i.e., under average), meaning that, to prevent the plants’ death, all 45 pots had to be regularly watered. Identical amounts of tap water were added to the (un)planted pots using a measuring cylinder. After 3, 6 and 12 months, respectively, 5 replicates of each modality were sacrificed for measurements. No sampling was performed in the winter because the plants would have slowed their activities. The PAHs’ residual and bioaccessible contents were determined in the soil samples. In the planted soil samples, the analyses of the PAHs were performed on rhizospheric soil. This was achieved by carefully removing plants from the cultured soil, shaking all soil particles that were coming off easily and then collecting soil that was close to the plant roots by gently scraping it off. The presence/absence of plants and their length from roots to shoots were noted on planted samples. Plants were then carefully washed and dried. Their fresh biomass was determined through weighing, then plants were dried at 40 °C for 48 h and their dry biomass was determined through weighing. The soil samples will be referred to according to the type (MS, TP or C) and the length of time (3, 6 and 12 months) of the treatment they received.

### 2.4. Chemical Analyses

#### 2.4.1. Dry Weight Determination

The soil samples’ dry weight determination was based on ISO 11465:1993 cor 1994 [23]. 

#### 2.4.2. Bioaccessible PAHs Determination in Soil Samples

The bioaccessible PAH determination in the soil samples was realized on fresh (i.e., freshly sampled and undried) soil samples, as described in the PAHs desorption kinetics section. The time of contact between the soil and Tenax^®^ beads through the aqueous solution was 24 h (see the PAHs desorption parameters paragraph of the results section for time choice).

#### 2.4.3. Total PAHs Determination in Soil Samples

The total PAH determination in the soil samples was based on ISO 13877:1998 [24]. The soils were chemically dried with an equivalent amount of anhydrous Na_2_SO_4_ and homogenized using a pestle and a mortar. The mixture was extracted for 16 h with dichloromethane on a Soxhlet device. The resulting organic phase was filtered on anhydrous Na_2_SO_4_, eliminated with a rotative evaporation device and replaced with n-hexane. Then, the extract was purified on basic Al_2_O_3_ before n-hexane was eliminated and replaced by acetonitrile. The final acetonitrile extract was weighed for volume determination and analyzed for PAHs.

#### 2.4.4. PAHs Analysis

The PAHs were analyzed in acetonitrile extracts of desorption kinetics, bioaccessible and residual samples according to ISO 13877:1998 [24]. Briefly 20 µL of PAHs in acetonitrile extract were injected on an Agilent reverse-phase C18 column (Eclipse PAH 4.6 × 250 mm, 5 µm) and eluted using acetonitrile and water, both acidified with formic acid (0.1% *v*/*v*). The elution flow rate was 1.5 mL min^−1^ and the acetonitrile/water gradient was: a linear increase from 50:50 to 75:25 from 0 to 15 min; a linear increase from 75:25 to 100:0 from 15 to 20 min; a 100:0 plateau from 20 to 40 min; and, finally, a linear decrease from 100:0 to 50:50 from 40 to 40.1 min, with a final isocratic hold of 2 min. The PAHs were detected fluorimetrically according to ISO 13877:1998 [24] and their quantification was achieved using external standard calibration.

### 2.5. Statistics

All statistical analyses related to the rhizoremediation experiment were carried out using Minitab 18.0. The equality of variances were verified according to Levene’s test, the data were analyzed by a general linear model or one-way analysis of variance, and mean values were compared by Tukey’s test at the 5% confidence level.

## 3. Results

### 3.1. PAHs Bioaccessibility Measurement

#### 3.1.1. Modelling PAHs Desorption Kinetics

After the soil samples were shaken for 1, 2, 4, 8, 16, 24, 48, 72 and 96 h in the presence of Tenax^®^ beads, the PAH fractions that remained sorbed to the soil were calculated according to Equation (1). Then, four desorption models (Table 2) were fitted on each PAH desorption dataset and on the desorption data for each group of PAHs (∑2–3 rings, ∑4 rings, ∑4–6 rings and ∑all). Afterwards, the BIC values were calculated using R for each model of each dataset. As explained previously, the objective was to select one model that would best describe the datasets. Thus, the BIC values were used to choose the best-fitted model for each PAH and are available in Table A1. The site distribution model had the smallest BIC value for the most individual PAHs, except for Fle, Anthr and Chrys, and for each group of PAHs, except for the ∑2–3 rings group. In three cases the first-order three-compartment model obtained the smallest BIC values, and in one case it was the first-order two-compartment model that obtained the smallest BIC value. However, each time the BIC values were three of four units lower than the BIC values of the site distribution model. This means that the supplement of information brought by the first order three-compartment (or two-compartment) model is “positive but not strong” compared to the site distribution model [25]. Thus, the site distribution model was chosen to describe all individual and groups of PAHs’ desorption data (Figure 1) and to calculate the *t_ex_* values. The parameters of the other models are not presented since they were not used afterwards.

#### 3.1.2. PAHs Desorption Parameters

The site distribution models’ parameters (alpha and beta) are presented in Table 3 along with the *t_ex_* values, calculated according to Equation (3). The alpha values ranged from 4.40 × 10^−4^ to 4.41 × 10^−3^, the beta values ranged from 2.17 × 10^−7^ h to 1.86 h, and the calculated extraction times were of 24 h for each compound and each group of PAHs. Therefore, a 24 h extraction time was used to determine the PAHs’ bioaccessible contents in the rhizoremediation experiment. As a comparison, when the PAH desorption kinetics were modelled on a different PAH aged-contaminated soils [15] the common extraction time was 48 h.

### 3.2. PAHs Rhizoremediation

#### 3.2.1. Plant Biomass

All plants’ seeds germinated well, which was expected given the 100% germination rate measured prior to the experiment and the fact that germination mobilizes a seed’s endosperm reserves [26]. However at the end of each culture period, the presence or absence of plants in each pot was noted along with their length from roots to shoots. Throughout the experiment, and despite good germination, TP plants never developed well, especially compared to MS which developed dense root systems. The plants in one pot were dead in the TP_3, MS_6 and MS_12 samples and the plants of three pots were dead in the TP_12 samples at the end of their respective culture period. Statistical analyses on the plants’ dry weights were performed after square root transformation. An analysis of variance showed significant interactions between time (3, 6, or 12 months) and treatment (C, MS, or TP). The results show that MS plants developed statistically more biomass than the TP plants as soon as after three months, and at the end of each culture period (Figure 2). 

#### 3.2.2. PAHs’ Bioaccessible and Residual Contents

Figure 3, Figure 4, Figure 5 and Figure 6 show the (un)planted soil samples residual and bioaccessible PAH contents at different stages of the rhizoremediation experiment. The presented data focusses on the different groups of PAHs (∑2–3 rings in Figure 3, ∑4 rings in Figure 4, ∑4–6 rings in Figure 5 and ∑all in Figure 6) as they summarize and emphasize the observations made on individual PAHs data. Statistical analyses on the residual contents were performed after log10 transformation. An analysis of variance showed significant interactions between time (3, 6, or 12 months) and treatment (C, MS, or TP) on both the residual and bioaccessible contents.

In all figures, the first obvious observation is that all groups of PAHs’ residual contents (Figure 3a, Figure 4a, Figure 5a and Figure 6a) exhibited similar patterns within each type of treatment. The residual PAH contents in C samples significantly diminished throughout the whole experiment and the samples reached about 8% of the total initial content (∑all) after 12 months. On the other hand, the residual contents in the TP samples diminished rather abruptly after 3 months of culture to about 10% of the total initial content, then remain statistically similar after 6 and 12 months. The most surprising pattern was exhibited by the MS samples’ residual contents. During the first 6 months, all the PAHs’ residual contents was lowered to about 10% of their initial content. After 12 months, the ∑2–3 rings content was statistically higher than after 6 months, and the other groups of PAHs’ residual contents clearly were not as low as after 6 months. The residual contents in the MS samples after 12 months were about 50% of the total initial content.

When it comes to the bioaccessible PAH contents (Figure 3b, Figure 4b, Figure 5b and Figure 6b), different observations can be made, and, this time, the patterns were different between the PAH groups. First, the ∑4–6 rings and ∑all contents did not significantly differ with treatment nor time, suggesting that, whilst the residual content globally lowers in all samples, bioaccessibility remains similar. When it comes to the ∑2–3 rings and ∑4 rings bioaccessible contents, the statistical analysis shows that they increased with time but in a more significant way in C samples.

## 4. Discussion

The objectives of this study were: (i) to adapt a comparative bioaccessibility measurement protocol using Tenax^®^ beads to an aged-contaminated soil, and (ii) to follow the PAHs’ bioaccessibility and residual contents in this soil in the presence of *Medicago sativa* L. or *Trifolium pratense* L., compared to unplanted control soil. The underlying hypothesis was that the continuous input of plant root exudates in situ could, as part of the rhizospheric effect, enhance the PAHs’ bioaccessibility, and thereby render them more susceptible to biodegradation by soil microorganisms.

Desorption kinetics were measured and modelled for 15 PAHs individually and grouped in categories (light, intermediate, heavy, and total). Out of the four models, the site distribution model was chosen to calculate the minimal common Tenax^®^ beads’ extraction time (24 h). During the course of the experiment (April 2018 to April 2019), an ISO norm to “determine the potential and environmental availability of a contaminant” was published (ISO/TS 16751:2018) [27]. The Tenax^®^ beads extraction protocol is overall similar to the one developed in this study and recommends a 20 h time of extraction, which is slightly less than calculated in this case. So even if the study was not conducted following a norm that came out while the study was ongoing, the protocols are very similar. Besides, the main objective of this protocol was to compare the bioaccessibility contents of soil that was submitted to different treatments, which was achieved.

Regarding the rhizoremediation experiment, it was conducted in the expectation of obtaining better PAH dissipation results. First of all, the *T. pratense* plants did not grow or last well in the experimental soil (Figure 2), even though the *T. pratense* seeds germinated well, as previously mentioned. Secondly, and even if some of the *M. sativa* plants died during the experiment, they developed more biomass than *T. pratense*. These outcomes were compared to a few results previously reported in the literature and summarized in Table 4. The presented results show similarities in the way that PAHs do not seem to affect germination but can affect growth by either decreasing it or increasing it. Importantly, Smith et al. [28] reported that *T. pratense* growth reduction could not have been foreseen by a traditional germination test. Therefore, the elevated amount of PAHs, although weathered, present in our experimental soil is likely to be responsible for the *T. pratense* plants decay in the long term.

The PAH levels were probably not the only factor that influenced the tested plants’ growth. Indeed, brownfield soils that are in need for remediation rarely present with a single type of contamination, and it has been pointed out that the experimental soil had Cr, Cu, Hg, Pb, and Zn contaminations. Besides, the cultures were conducted on a very high pH (10.0). To the best of our knowledge, it is difficult to know whether *M. sativa* or *T. pratense* are tolerant to such elevated pH since this value is out of the usual working range encountered in traditional soil use, such as agriculture, and there is no information on that matter in the literature. The choice not to use amendments in this experiment originates from economic considerations. Indeed, many brownfields already lack management and remediation because of financial considerations. Some brownfields are considered worth the remediation investment, some are not. So, the experimental setup aimed at exploring and developing remediation techniques that are as low-cost and low-maintenance as possible, hence the initial choice to not use amendments. However, such growth conditions might have caused some of the plants to decay since an elevated pH lowers essential nutrient availability in soil solution [32]. Therefore, it would be interesting to repeat this rhizoremediation experiment by amending the soil with plant essential nutrients to enhance their growth (especially *T. pratense* L. in this case). But it is important to emphasize that the use of such soil in the experiment provided observations as to *M. sativa*’s and *T. pratense*’s capacity to enhance rhizoremediation in realistic and unoptimized conditions, and shows that there is still some research that needs to be conducted on the phytoremedation of soils presenting multiple types of contaminations. Finally, a possible explanation as to why *M. sativa* plants were less affected than *T. pratense* probably lies in the structural differences between the two tested plant species. Indeed *M. sativa* has a deep taproot which is a great adaptation to sandy soils, whereas *T. pratense* has a shallow and highly branched root system, which is not as efficient on a more sandy soil such as the experimental soil (which, as a reminder, identified as loamy sand).

Regarding the PAH residual contents from the rhizoremediation experiment, contrasting results have already been published in the literature and are summarized in Table 5. We should, however, mention that there were many more phytoremediation assays involving *M. sativa* L. than *T. pratense* L. Besides, many studies involving PAH phytoremediation were either performed on soil freshly spiked with PAHs (which often are only a few representative compounds such as BaA, Pyr or Phen), or on spiked soil that was allowed to age for a few weeks, and sometimes a few months. Fewer studies were performed on aged-contaminated soil such as the one used in this study, and, if such soil was experimented with, the growing conditions were controlled as trials often took place in greenhouses, or the initial PAH concentrations were sometimes much lower than for our tested experimental soil. Also, some of them lacked unplanted control to compare the PAH dissipation results. The experiment by Olson et al. [33] is the most similar to the one in this study in terms of the PAHs’ diversity, initial content, and final dissipation rates compared to an unplanted control. The authors hypothesized that the symbiosis relationship of the *Fabaceae* plants with rhizobia offered long-term advantages to the plants and their rhizosphere microbial community but have not observed a correlated raise in the PAH-degrading microbial community to corroborate their hypothesis.

However, and concerning the PAHs’ residual contents presented in this study, several hypotheses were formulated to explain the unexpected fact that MS_12 residual contents were higher than the MS_6 contents. (i) The easiest would be to acknowledge the large natural variability of biological experiments. As a reminder, samples were sacrificed at the end of each culture period so data from increasing time periods do not represent the continuity of the same planted pots, meaning either the MS_6 or the MS_12 samples data could constitute an exception. But, since MS plants were statistically as developed after 12 months as after 6 months (Figure 2), similar (or lower) PAHs’ residual contents were expected to be measured at the end of the experiment. (ii) PAHs could have been temporarily sequestered by plants and then released through roots decay. PAHs can be adsorbed onto the root cell membranes, as was reported for naphthalene with *M. sativa* roots by Schwab et al. [37] and for phenanthrene and pyrene with *Lolium multiflorum* Lam. by Kang et al. [38], who both concluded that the adsorbed amounts were linked to cell lipid contents. Besides, the fine roots of perennial plants continuously grow and die over time [39], with periods of either net production or net loss throughout the year, suggesting PAHs could have been released back to the soil because of root decay taking place during the second part of the experiment, which corresponds to the end of autumn and winter. (iii) Given the dry culture conditions (several droughts combined to a sandy draining soil) that plants endured, and the high capacity of *M. sativa* L. to draw water with dense and deep root systems, PAHs might actually have dissipated from the plants rhizosphere in MS samples (either by volatilization, degradation, or lixiviation) during the first 6 months, and, as plants roots grew denser, they might have vertically reached and retained more PAHs in their rhizosphere.

To summarize the PAH residual contents results, it can be stated that: (i) in a short time (3 months) the presence of *T. pratense* L. plants led to greater PAHs’ dissipation than in the control and *M. sativa* L. samples, which tend to confirm *T. pratense* L.’s potential for phytoremediation, whilst *M. sativa* L. did not enhance PAHs dissipation compared to control samples. (ii) Dissipation in *T. pratense* L. samples was similar after 3, 6 and 12 months, regardless of the fact that many *T. pratense* L. samples plants died during the experimental period. (iii) After 6 months, dissipation in control samples was statistically similar to dissipation in planted samples, which was confirmed in the long-term (12 months) for *T. pratense* L. samples but not for *M. sativa* L. samples, which presented higher residual contents. If PAHs were dissipated through biodegradation mechanisms, it would mean that plants did not enhance biodegradation in the long term. However, if dissipation simply results from leaching and/or lixiviation, the slower dissipation in presence of *M. sativa* L. could be caused by roots preventing vertical migration by physically retaining soil particles or “pumping up” contaminated soil solution, which would be confirmed by the MS_12 residual contents. All the mentioned hypotheses could be investigated by repeating this experiment for another year, comparing data, and analyzing plants’ PAH contents after shorter culture periods, (i.e., every month for a year) to follow more accurately the fate of PAHs in the presence of these plants.

The PAHs’ bioaccessible results were compared to previously published information summarized in Table 6. References were chosen that presented various tested remediation techniques, similar bioaccessibility measurement protocols, and, of course, were performed on aged-contaminated soils. The reported remediation techniques in Table 6 are either phytoremediation, biostimulation (which enhances existing microorganisms’ activity through the use of amendments or optimized conditions) that were applied through biopiles or composting, bioaugmentation (which inoculates specialized degrading strains to a soil), and chemical oxidation. The results vary in terms of residual PAHs’ diminution, but these concentrations always decrease or remain similar. Also, lighter PAHs’ contents (such as Phen) tend to decrease more than heavier PAHs’ contents (such as BaA), which was not observed in the present experimental results. The bioaccessible PAHs’ contents, however, show contrasting patterns. Posada-Baquero et al. [40,41] reported that, generally speaking, techniques such as phytoremediation or biostimulation seem to lead to decreases in PAHs bioaccessible contents, whilst techniques that were more focused on influencing bioaccessibility, such as the addition of surfactants or bioaugmentation, seem to lead to increases in the PAHs’ bioaccessible contents. However, the results reported by Medina et al. [42] also showed an increase in PAHs bioaccessible contents after biostimulation was employed. A similar pattern was observed after chemical oxidation. The results presented in this paper are also in contradiction with the theory exposed by Posada-Baquero et al. [40], even though the reported phytoremediation results are based on different plants. In the present paper, the PAHs’ bioaccessible contents throughout the rhizoremediation trial show almost similar patterns for (un)planted soil samples (Figure 3b, Figure 4b, Figure 5b and Figure 6b). The light and intermediate PAHs’ bioaccessibility raised throughout the experiment but globally (∑all) remained unchanged. This suggests that there is no global effect of *M. sativa* L. nor *T. pratense* L. culture on bioaccessibility, which would mean that equilibrium balances unrelated to the plants presence or absence are filling the vacancy left by the dissipation of PAHs throughout the experiment. Therefore, it would seem reasonable to conclude that the tested *Fabaceae* do not enhance the PAHs’ bioaccessibility compared to unplanted soil. However, the less significant rise of the light and intermediate PAHs’ bioaccessibility in planted samples compared to control samples suggests that the *M. sativa* L. or *T. pratense* L. plants’ presence actually slows the increase of bioaccessibility. According to Ouvrard et al. [10], this would make sense considering that PAHs are hydrophobic compounds that tend to sorb on organic soil content, and part of the PAHs released in soil aqueous solution could have been sorbed onto plants exudates, explaining a less important increase of bioaccessibility in planted samples. The lowering of bioaccessibility might also be caused by interactions between the targeted pollutants and some surface-active compounds released from the plant roots into the rhizosphere. It has indeed been demonstrated that surface-active compounds (such as saponins) could form micelles that can enhance the PAHs apparent solubility in the environment [43]. However, it has also been demonstrated that hydrophobic interactions can take place between soil particles and the surfactants [44], meaning PAHs could be partitioned into micelles or hemimicelles bound to hydrophobic constituents of the rhizosphere, such as soil particles or even lipid membranes from the roots. A similar hypothesis has already been advanced in a previous study that aimed to increase PAHs’ apparent solubility by washing an aged-contaminated soil with aqueous solutions of saponins from *Quillaja saponaria* Molina bark [45]. The results showed a less efficient extraction of PAHs if the surfactant concentration was too elevated. Such seclusion of PAHs away from biodegradation agents would thus explain why *M. sativa* L. and *T. pratense* L. presence in soil lowered the pollutants bioaccessibility instead of increasing it.

## 5. Conclusions

As a conclusion, the general rhizoremediation results suggest that when plants are small or absent, the PAHs’ residual contents seem to globally dissipate faster from the rhizosphere and the bioaccessibility contents to increase a little faster (at least for light and intermediate PAHs). From a remediation point of view, it means this one-year trial brought no evidence that the presence of *M. sativa* L. or *T. pratense* L. on this aged-contaminated soil was beneficial on the PAHs’ remediation process, compared to unplanted soil. However, from an environmental risk point of view, the slower dissipation but also the bioaccessibility enhancement of PAHs in the presence of those plants could be used as a tool to prevent the migration of the contaminants towards more sensitive environmental compartments such as ground or even surface water.

Let us also point out here that contrasting observations have been made previously in the literature. First, as to the PAHs effects on *M. sativa* L. and *T. pratense* L. growth, whether cultures took place on freshly spiked or aged-contaminated soil, but also as to the effect of those plant types on PAHs remediation, this study added information to previously acquired data. Because it led to mitigated conclusions, it highlights the complexity of plant-soil-pollutant interactions and the fact that there might be antagonist events taking place within this system. It also points out the need to perform more phytoremediation experiments on a broad range of aged-contaminated soils types presenting with different pedologic characteristics and different levels and types of contamination to try and predict the conditions in which plants might grow and enhance PAHs’ remediation. Also, it points out the importance of a thoughtful selection of the plants to try and remediate the contaminated soils, as they are likely to be confronted with difficult growth conditions such as extreme pH, poor nutrient availability or inadequate soil drainage. Finally, we would like to insist that the parallel evaluation of both the PAHs’ bioaccessible and residual contents, as was performed in this study, could bring new insights to the complexity of soil remediation trials in general, if they were to be realized more systematically.

## Figures and Tables

**Figure 1 ijerph-17-04016-f001:**
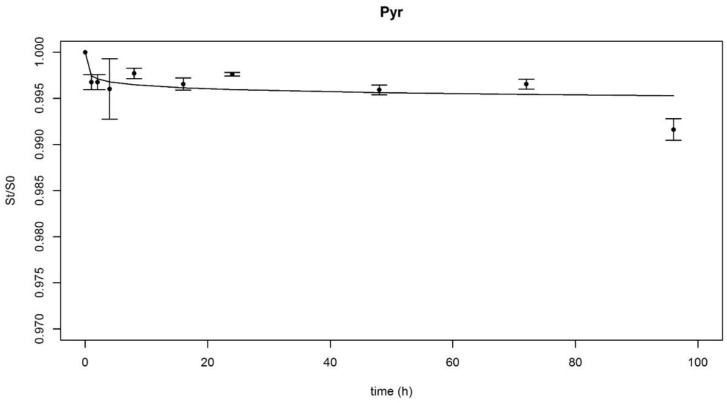
Example of desorption kinetic obtained using Tenax^®^ beads (the data and the modelling are for pyrene). St/S0 is the remaining sorbed fraction according to extraction time. The dots are the data means ± confidence interval (α = 5%, *n* = 4 or 5); the line is the fitted site distribution model.

**Figure 2 ijerph-17-04016-f002:**
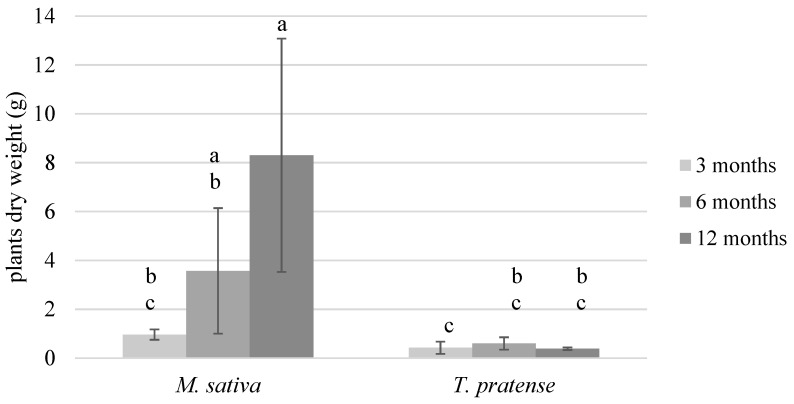
Plants dry weight (biomass) after each culture period. Within each group, bars that share the same letter are not significantly different (*p* > 0.05). The values are means ± confidence interval (α = 5%; *n* = 5; *n* = 4 for TP_3, MS_6 and MS_12; and *n* = 2 for TP_12).

**Figure 3 ijerph-17-04016-f003:**
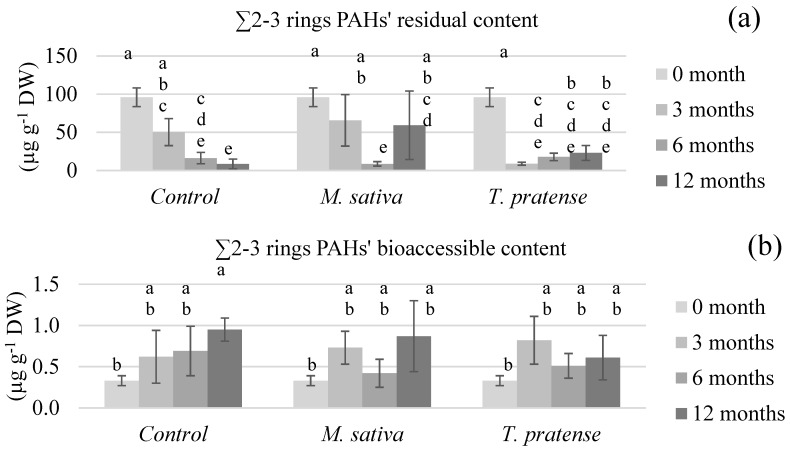
The light PAHs’ (∑2–3 rings) (**a**) residual and (**b**) bioaccessible content of soils planted with *M. sativa* L. or T. *pratense* L. compared to unplanted control samples after different time periods. The values are means ± confidence interval (α = 5%, *n* = 5). There is a significant interaction between the type and the time of culture, so within each PAH fraction (residual or bioaccessible) sticks that share the same letter are not significantly different (*p* > 0.05).

**Figure 4 ijerph-17-04016-f004:**
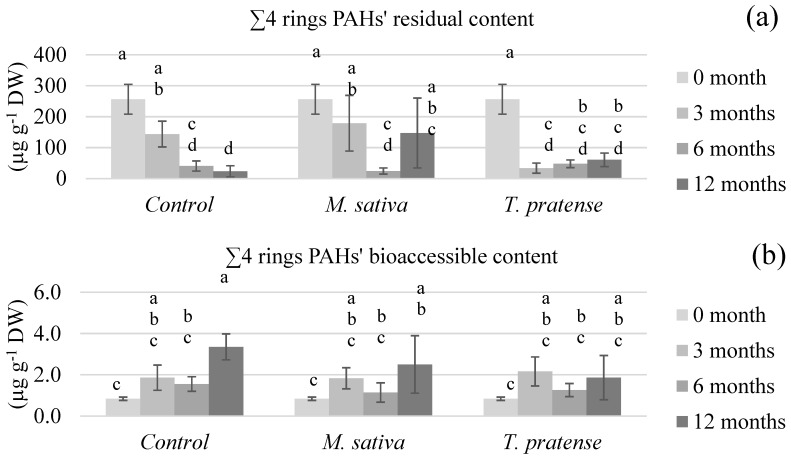
The intermediate PAHs’ (∑4 rings) (**a**) residual and (**b**) bioaccessible content of soils planted with *M. sativa* L. or *T. pratense* L. compared to unplanted control samples after different time periods. The values are means ± confidence interval (α = 5%, *n* = 5). There is a significant interaction between the type and the time of culture, so within each PAH fraction (residual or bioaccessible) sticks that share the same letter are not significantly different (*p* > 0.05).

**Figure 5 ijerph-17-04016-f005:**
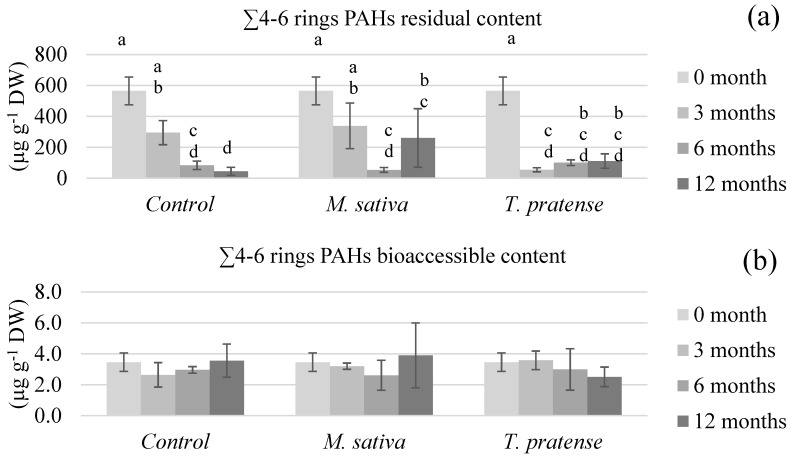
The heavy PAHs’ (∑4–6 rings) (**a**) residual and (**b**) bioaccessible of soils planted with *M. sativa* L. or *T. pratense* L. compared to unplanted control samples after different time periods. The values are means ± confidence interval (α = 5%, *n* = 5). There is a significant interaction between the type and the time of culture, so within each PAH fraction (residual or bioaccessible) sticks that share the same letter are not significantly different (*p* > 0.05).

**Figure 6 ijerph-17-04016-f006:**
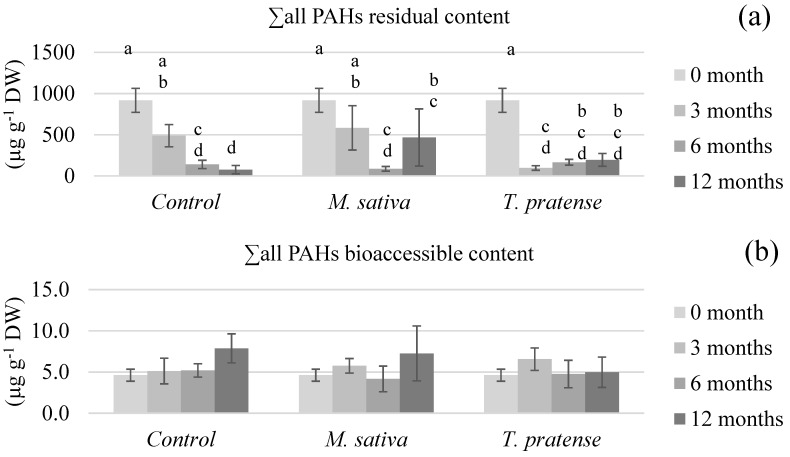
The total PAHs’ (∑all) (**a**) residual and (**b**) bioaccessible contents of soils planted with *M. sativa* L. or *T. pratense* L. compared to unplanted control samples after different time periods. The values are means ± confidence interval (α = 5%, *n* = 5). There is a significant interaction between the type and the time of culture, so within each PAH fraction (residual or bioaccessible) sticks that share the same letter are not significantly different (*p* > 0.05).

**Table 1 ijerph-17-04016-t001:** Experimental soil polycyclic aromatic hydrocarbons’ (PAHs) initial contents.

PAHs	Abbreviation	µg g^−1^ DW
Naphthalene	N	20.2 ± 2.4
Acenaphthene	Ace	1.0 ± 0.4
Fluorene	Fle	5.1 ± 0.9
Phenanthrene	Phen	45.5 ± 7.2
Anthracene	Anthr	24.1 ± 3.6
Light PAHs	∑2–3 rings	95.9 ± 12.2
Fluoranthene	F	139 ± 36.6
Pyrene	Pyr	117 ± 20.5
Intermediate PAHs	∑4 rings	256 ± 47.9
Benzo(a)anthracene	BaA	79.2 ± 10.5
Chrysene	Chrys	73.6 ± 8.5
Benzo(b)fluoranthene	BbF	96.0 ± 19.4
Benzo(k)fluoranthene	BkF	48.1 ± 5.0
Benzo(a)pyrene	BaP	95.2 ± 15.6
Dibenzo(ah)anthracene	DBahA	12.1 ± 1.3
Benzo(ghi)perylene	BghiP	66.3 ± 25.3
Indeno(123-c,d)pyrene	IcdP	94.3 ± 21.7
Heavy PAHs	∑4–6 rings	565 ± 90.0
Total PAHs	∑all	917 ± 146

Values are mean ± confidence interval (α = 5%, *n* = 5).

**Table 2 ijerph-17-04016-t002:** Desorption models tested to describe the measured desorption kinetics of PAHs in the experimental soil. Models were adjusted using the Levenberg–Marquardt algorithm [22].

**First-Order Model**	**(1 Parameter)**
$\frac{S_{t}}{S_{0}}=e^{-kt}$	
**First-Order Two-Compartment Model**	**(4 Parameters)**
$\frac{S_{t}}{S_{0}}=F_{rap}\times e^{-k_{rap}t}+F_{slow}\times e^{-k_{slow}t}$ $F_{rap}+F_{slow}=1$	
**First-Order Three-Compartment Model**	**(6 Parameters)**
$\frac{S_{t}}{S_{0}}=F_{rap}\times e^{-k_{rap}t}+F_{int}\times e^{-k_{int}t}+F_{slow}\times e^{-k_{slow}t}$ $F_{rap}+F_{int}+F_{slow}=1$	
**Site Distribution Model**	**(2 Parameters)**
$\frac{S_{t}}{S_{0}}={\left(\frac{\beta }{\beta +t}\right)}^{\alpha }$	

**Table 3 ijerph-17-04016-t003:** Fitted parameters of the site distribution model for the different PAHs and *t_ex_* values calculated according to Equation (3).

	β (h)	α (-)	*t_ex_* (h)
N	1.86 × 10^0^	4.41 × 10^−3^	24
Ace	7.30 × 10^−2^	3.72 × 10^−3^	24
Fle	1.06 × 10^−1^	2.05 × 10^−3^	24
Phen	4.39 × 10^−2^	1.33 × 10^−3^	24
Anthr	6.94 × 10^−2^	1.53 × 10^−3^	24
F	1.09 × 10^−1^	1.37 × 10^−3^	24
Pyr	4.92 × 10^−3^	4.77 × 10^−4^	24
BaA	8.84 × 10^−2^	1.62 × 10^−3^	24
Chrys	1.32 × 10^−1^	2.20 × 10^−3^	24
BbF	6.57 × 10^−3^	1.04 × 10^−3^	24
BkF	2.13 × 10^−2^	1.41 × 10^−3^	24
BaP	6.09 × 10^−4^	6.84 × 10^−4^	24
DBahA	2.17 × 10^−7^	4.40 × 10^−4^	24
BghiP	5.93 × 10^−7^	5.38 × 10^−4^	24
IcdP	6.73 × 10^−5^	5.61 × 10^−4^	24
∑2–3 rings	2.47 × 10^−1^	1.95 × 10^−3^	24
∑4 rings	4.97 × 10^−2^	9.47 × 10^−4^	24
∑4–6 rings	6.84 × 10^−3^	1.07 × 10^−3^	24
∑all	1.84 × 10^−2^	1.11 × 10^−3^	24

**Table 4 ijerph-17-04016-t004:** Comparison of germination and growing conditions and outcomes between a few published references and the presented experimental soil, for *Medicago sativa* L. and *Trifolium pratense* L.

Reference	Tested Plant (s)	Germination/Growing Conditions	Germination/Growing Outcomes	Germination/Growing Conditions in Presented Experimental Soil (Table 1)
Sverdrup et al. [29]	*T. pratense* L.	Soil freshly spiked with Fle, Phen, F, and Pyr at individual concentrations up to 1000 mg kg^−1^ DW	No seed emergence inhibition; 20% plant growth inhibition starting at concentrations: 55 mg kg^−1^ DW (Fle), 37 mg kg^−1^ DW (Phen), 140 mg kg^−1^ DW (F), and 49 mg kg^−1^ DW (Pyr).	Individual concentrations of: 20.2 mg kg^−1^ DW (N); 5.1 mg kg^−1^ DW (Fle); 45.5 mg kg^−1^ DW (Phen); 139 mg kg^−1^ DW (F); 117 mg kg^−1^ DW (Pyr); 79.2 mg kg^−1^ DW (BaA); 73.6 mg kg^−1^ DW (Chrys). The total concentration of 15 PAHs was 917 mg kg^−1^ DW.
Smith et al. [28]	*T. pratense* L.	Soil spiked with seven PAHs and aged for 4 weeks (total concentration was 450 mg kg^−1^ DW after the ageing process)	Germination was not affected; growth was significantly reduced (70%).	
		Aged-contaminated soil (total concentration of 16 PAHs was 5300 mg kg^−1^ DW)	Germination was not affected; growth was significantly reduced (65%).	
Henner et al. [30]	*M. sativa* L.; *T. pratense* L.	Pure saturated solutions of N, Phen, F, Chrys and BaA	Similar germination levels as in the absence of PAHs.	
		Aged-contaminated soil (total concentration of 16 PAHs was 1500 mg kg^−1^ DW)	Germination slowed (3–4 days) but reached similar levels as in uncontaminated soil; plant growth was inhibited (80%) for *M. sativa*; No information for *T. pratense*.	
Afegbua and Batty [31]	*M. sativa* L.	Soil spiked with Phen (300 mg kg^−1^ DW), F (200 mg kg^−1^ DW), and BaA (5 mg kg^−1^ DW) then aged for 4 weeks.	Shoots and roots dry biomass respectively increased by 110% and 40% when PAHs were mixed.	

**Table 5 ijerph-17-04016-t005:** Comparison of phytoremediation conditions and outcomes between a few published references for *Medicago sativa* L. and *Trifolium pratense* L.

Reference	Tested Plant (s)	Phytoremediation Conditions	Phytoremediation Outcomes
Fan et al. [34]	*M. sativa* L.	Soil freshly spiked with Pyr (500 mg kg^−1^ DW).	6% better removal in the rhizosphere compared to the non-rhizosphere soil.
Hamdi et al. [35]	*M. sativa* L.	Soil spiked with BaA (100 mg kg^−1^ DW) + 15-month landfarming (bioremediation process) had brought content down to 9 mg kg^−1^ DW. Then soil was planted 5 months in controlled conditions.	BaA content lowered to 4.3 mg kg^−1^ DW. No unplanted control to compare results.
Teng et al. [36]	*M. sativa* L.	Agricultural weathered soil (total concentration of 16 PAHs was 10 mg kg^−1^ DW) was planted for 3 months.	45% lowering of the 16 PAHs mixture.
Olson et al. [33]	*M. sativa* L.; *T. pratense* L.	Weathered soil (total concentration of 17 PAHs was 753 mg kg^−1^ DW) was planted 14 months in controlled conditions.	Total PAHs dissipation was not different from unplanted control samples, after 7 and 14 months.

**Table 6 ijerph-17-04016-t006:** Comparison of remediation conditions and outcomes between a few published references for PAHs residual and bioaccessible contents.

Reference	Initial Soil Concentrations	Remediation Conditions	PAHs Residual Concentrations Evolution	PAHs Bioaccessible Concentrations Evolution
Posada-Baquero et al. [40]	Phen and BaA concentrations were 843.10 and 56.5 mg kg^−1^, respectively; Phen and BaA bioaccessible concentrations were 0.75 and 0.10 mg kg^−1^, respectively.	5 months biostimulation in a biopile amended with urea and KH_2_PO_4_; No reported control.	Phen diminished by over 94%; BaA diminished by about 35%.	Phen diminished by almost 90%; BaA diminished by 30%.
	Phen and BaA concentrations were 197.10 and 4.12 mg kg^−1^, respectively; Phen and BaA bioaccessible concentrations were 0.42 and 0.20 mg kg^−1^, respectively.	60 days sunflowers phytoremediation in a greenhouse; No reported control.	Phen diminished by over 97%; BaA diminished by about 46%.	Phen diminished by over 86%; BaA diminished by 70%.
	Phen and BaA concentrations were 36.7 and 0.64 mg kg^−1^, respectively; Phen and BaA bioaccessible concentrations were 0.23 and 0.03 mg kg^−1^, respectively.	60 days bioaugmentation with specialized strains; No reported control.	Phen diminished by over 30%; BaA diminished by over 10%.	Phen raised by over 140%; BaA raised by 300%.
	Phen and BaA concentrations were 46.3 and 1.40 mg kg^−1^, respectively; Phen and BaA bioaccessible concentrations were 0.27 and 0.024 mg kg^−1^, respectively.	60 days bioaugmentation with specialized strains; No reported control.	Phen diminished by 60%; BaA did not diminish.	Phen raised by over 35%; BaA raised by over 200%.
Medina et al. [42]	Aged-contaminated soil (PAHs concentration was 214 mg kg^−1^ and bioaccessible PAHs fraction was 1%).	Chemical oxidation with ammonium persulfate; No reported control.	PAHs diminished by almost 30%.	PAHs raised to a 19% fraction of remaining total PAHs.
	Aged-contaminated soil (PAHs concentration was 151 mg kg^−1^ and bioaccessible PAHs fraction was 19%).	12 months incubation (served as control).	PAHs diminished by 25%.	PAHs raised to a 30% fraction of remaining total PAHs.
		12 months biostimulation through composting with amended goat manure.	PAHs diminished by 33%.	PAHs raised to a 56% fraction of remaining total PAHs.
Posada-Baquero et al. [41]	Aged-contaminated soil (PAHs concentration was 513 mg kg^−1^ and bioaccessible PAHs fraction were 60 and 40% for light and heavy PAHs, respectively).	210 days of sunflower phytoremediation in a greenhouse combined to a biosurfactant amendment after 75 days.	Light and heavy PAHs diminished by over 90% and 70% in (un)planted soil samples, respectively; Biosurfactant addition had no effect.	Light and heavy PAHs diminished under 10 and around 10% in (un)planted soil samples, respectively; Biosurfactant addition enhanced all PAHs bioaccessible fractions in planted samples for a few days; At the end, bioaccessible fractions were similar in all samples.

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
