# Peer review of "Temporal Evolution of PAHs Bioaccessibility in an Aged-Contaminated Soil during the Growth of Two Fabaceae"

_ijerph, 2020, doi:10.3390/ijerph17114016_

Round 1

Reviewer 1 Report

The manuscript is prepared very well. Methodology and analysis of results don't raise any objections. I think that the reason for so weak growth of. T. pretense was this pH value. And, I agree that you should repeat in the future this experiment by adding f.ex. some lime. To use all the beneficial effects of rhizoremediation it is vital to create the right conditions  for the plants to growth.

I have only some suggestions connected with edition of the manuscript:

Page 1, lines 39 – 40 “Because…. “and then “PAHs have been…” Shouldn’t these two sentences be as one sentence?

Page 6 – lines:202 – 213 – is it necessary in the text of publication? I guess this is a mistake.

I  hope that my comments will be helpful in improving this manuscript.

Reviewer 2 Report

Hi,

This work tests the capabilities of 2 plants (M. sativa and T. pratense) to reduce the bioavailability of PAHs in a polluted over the course of one year (april’18-19). The experimental work is well conducted, the work is well written in a very well written English and it is highly noted that the authors have experience with phytoremediation studies of this sort. This is a typical work of minor revision since it is very consistent, but I have some concerning aspects that are difficult to solve.

First, the work do not present many novelties in the methodology. As strong points, the study of bioavailability in organic pollutants is less frequent than the classic PTEs, and the use of Bayesian methods gives the work entity. On the other hand, the study of phytoremediation for remediate soils polluted by PAHs, and more specially, with the plants chosen, is widely studied. Something that the authors themselves declare in lines 370-372.

Another important issue is that authors declare that “This one-year trial brought no evidence that the presence of M. sativa or T. pratense on the aged contaminated soil was beneficial on the PAHs remediation process, compared to unplanted soil.”, thus declaring that the main objective of the work wasn’t achieved.

I wonder why authors didn’t consider the use of proven and rising amendments (biochar, compost, nanoparticles, etc) to enhance the yield of the plants. Their use is highly encouraged in the phytoremediation for the recovery of brownfields, and could even reduce the bioavailability of the pollutants, but these are not even recognized on the work, just mentioned in line 360 as a possibility. I think these methods should be reviewed somewhere in the paper.

Then, I have detected minor errors in the work.

For instance, lines 202-213 evidence a lack of reviewing prior sending. I don’t mind at all but please be careful with these things next time.

Figure 3 letters of abscissa axis are not readable in “a)”.

Lines 308-313. In my opinion, there is no need on remarking the objectives of the work again in such a direct manner in this point.

In general, I find the discussion too dense to read. The paragraphs are too long, and some things can be summarized into tables (370-4XX) to make the  discussion more enjoyable. Please, improve it.

In brief, when reading the work, I have the impression that authors have full knowledge of the working procedures, and that they have put love on it, but they have performed an experiment whose results were not the expected ones. This happens sometimes and it is a shame in long experiments like the one presented here. So whether the editor considers the results are enough and acceptable for the journal, despite the main hypothesis wasn't achieved, therefore I recommend what I’ve clicked, the minor revision.

Reviewer 3 Report

I think that this article has been a great effort because the experiment has been long and with a great number of samples to study and because the authors have studied bioaccessibility and residual concentration of PAHs. However, I suggest that the authors do major revisions of the manuscript before acceptance for publication, so specific comments can be find below:

  • Introduction: this section is well but I think that it will be necessary write some about the importance of to measure bioaccessibility or bioavailability in bioremediation and phytoremediation process. In addition, explain also some related to aging and how this effect can affect to bioaccessibility measurements is very important.

In this section in the line 88 the word untreated soil should be better explained because in this paper it may be confused by the readers with the word unplanted.

  • Materials and methods: when the sentence total PAHs is used, it is better write “sum of 16 EPA PAHs” or explain that PAHs recommended by EPA have been measured.

In the line 152 of this section the use of the sentence “uncontrolled conditions” needs to be better explained.

Explain also why these experimental sampling point have been selected

The final time of the experiment was chosen because it coincides with the completion of the plants cycle?

In the line 175 of this section the sentence “fresh soil” should be better explained because the readers probably don’t understand it.

  • Results: attending to the title of this paper, the most important results from this study have been represented in figure 3 and 4. The importance of study the evolution of bioaccessibility or bioavailability in an experiment is the effect of this in the total or residual concentration of PAHs in the soil. For this reason, it could be better represented in the same panel (figure) bioaccessbility and residual concentration. For example, they can represent on the left the residual concentration and on the right bioaccessible concentration for all the treatment study in this work. In another figure like the previous can be represent the same results for the control treatments. To do these recommendations it is better represented the figures using the option in excell of linear dispersion graphs (XY) and not using bars graphics.

With respect to results of PAHs desorption, in the results it is not very clear which option from table 2 are being used and in addition why the results of the different parameters of these model have not been represented in the results?? I ask it because for example to assess bioaccessibility it is very important the values of Frap or Fslow to understand what is happening during the experiment.

  • Conclusions: in general, from my point of view, this section is too long and it is not very clear. In addition, in this section the authors had done a lot of effort in discuss results about the plants and I think that if we attended to the title of this paper, they should do more emphasis in discuss results about bioaccessibility.

I suggest that the authors could compared data of biaccesibility obtained in the soil that they have used,with data from biaccessibility from others aged soil contaminated with PAHs which appear in different previous paper because I know that there are a lot of.

Round 2

Reviewer 3 Report

From my point of view the authors have responded favourably to all the reviewers' proposals and they have managed to improve the quality of this paper. 

I think that this paper could be accept in this new format.